# Significance of Melt Pool Structure on the Hydrogen Embrittlement Behavior of a Selective Laser-Melted 316L Austenitic Stainless Steel

**DOI:** 10.3390/ma16041741

**Published:** 2023-02-20

**Authors:** Jie Liu, Huajie Yang, Lingxiao Meng, Di Liu, Tianqi Xu, Daokui Xu, Xiaohong Shao, Chenwei Shao, Shujun Li, Peng Zhang, Zhefeng Zhang

**Affiliations:** 1Department of Materials Science and Engineering, Shenyang University of Chemical Technology, Shenyang 110142, China; 2Institute of Metal Research, Chinese Academy of Sciences, Shenyang 110016, China; 3Department of Materials Science and Engineering, University of Science and Technology of China, Shenyang 110016, China

**Keywords:** hydrogen embrittlement, selective laser melting, austenitic stainless steel, mechanical properties, quasi in situ EBSD

## Abstract

The hydrogen embrittlement (HE) behavior of a selective laser-melted (SLM) 316L austenitic stainless steel has been investigated by hydrogen charging experiments and slow strain rate tensile tests (SSRTs) at room temperature. The results revealed that compared to the samples without H, the ultimate tensile strength (UTS) and elongation (EL) of specimens were decreased from 572 MPa to 552 MPa and from 60% to 36%, respectively, after 4 h of electrochemical hydrogenation with a current density of 100 mA/cm^2^. The negative effects of hydrogen charging were more pronounced on the samples’ ductility than on their strength. A quasi in situ EBSD observation proved that there was little phase transformation in the samples but an increased density of low angle grain boundaries, after 4 h H charging. After strain was applied, the surface of the H-sample displayed many hydrogen-induced cracks along the melt pool boundaries (MPBs) showing that these MPBs were the preferred areas for the gathering and transferring of hydrogen.

## 1. Introduction

Additive manufacturing (AM) technology is a layer-by-layer technology based on the computer-aided design (CAD) system, which is also called 3D printing, represents a realistic alternative to many conventional manufacturing techniques [1,2,3,4]. As one of the most important AM technologies, selective laser melting (SLM) [5,6,7] uses high-power laser heat sources and powder raw materials that can rapidly fabricate metal components with high density and complex structures. 316L austenitic stainless steel (SS) is usually used in the medical industry, aerospace, transportation, food and other industries owing to its exceptional properties, i.e., mechanical, corrosion resistance, biocompatibility and low cost [8,9]. Due to the excellent corrosion resistance of 316L SS, it has been commonly used for hydrogen (H) storage vessels, hydrogen infrastructure, as well hydrogen cell applications [10,11]. 

Compared with the conventional produced 316L SS, the SLM benefits from fast efficiency to produce customized parts as well as heterogeneous microstructures that contain a molten pool, and fine cellular and columnar sub-structure [12,13]. Because of these advantages, the use of SLM 316L SS has gradually become an excellent choice in hydrogen environments. Therefore, with the increasing application of SLM technology in its challenging service environment, the resistance of SLM 316L SS to hydrogen embrittlement (HE) is crucial.

Recently, some studies have investigated the hydrogen behavior of selective laser-melted austenitic stainless steel. Khaleghifar et al. [14] found that SLM 316L SS has a slight reduction in elongation after H charging. Bertsch et al. [15] systematically investigated the influence of hydrogen on the mechanical properties of SLM 316L SS, indicating that the presence of hydrogen leads to decreasing plasticity. By comparing the HE susceptibility of SLM 304L SS (and casting plus annealing 304L SS), Lee et al. [10] reported that SLM 304L SS has a higher resistance to HE than the conventionally manufactured 304L SS. They found that the HE susceptibility of 304L SS varied with the austenite stability and H-trapping behavior. Compared with wrought 316L SS, Kong et al. [16] found that SLM 316L SS undergoes a small martensite phase transformation after hydrogen charging. Miller et al. [17] reported that the MPBs act as the trap sites for hydrogen, further causing brittle fracturing. Based on these contradictory results, it seems that the HE sensitivity of SLM stainless steel samples depends on both internal and external factors. For example, the microstructure and environmental conditions may be crucial factors for HE behavior and the mechanical properties of SLM samples. Recent studies demonstrate that different microstructures could significantly influence the HE behavior of the AM steels [18,19]. However, systematical studies on the relationship between HE behavior and the heterogenous microstructure of SLM 316L SS are few. It is crucial to reveal the HE behavior and HE resistance of SLM 316L SS to make it a candidate in hydrogen service conditions.

The overall aim of this work is to investigate the evolutions of microstructural features and tensile properties after hydrogen charging. This will provide some data support for the study of HE behavior of SLM 316L stainless steel. The slow-strain-tensile tests (SSRTs) with a low strain rate of 1 × 10^−5^ s^−1^ were used to quantitatively determine the effect of hydrogen on the tensile properties of specimens. Multiple electron microscopy techniques were used to analyze the microstructure characteristics and tensile deformation mechanisms of samples, with and without H, to extend our understanding of the HE behavior of SLM 316L SS.

## 2. Experimental Procedures

### 2.1. Material Fabrication

The material in this study is an SLM-316L SS sample, which was fabricated on an EOS M280 SLM machine under an argon atmosphere. The optimal processing parameters in this study are presented in Table 1. The nominal per cent elemental composition pf the weight of the SLM materials is given in Table 2. 

### 2.2. Electrochemical Hydrogen Charging and SSRT Test

Tensile specimens were cut along the building direction and the processing parameter for the samples are presented in Figure 1a. Before tensile testing, all samples were sequentially ground with 400# to 3000# sandpaper, polished to mirror brightness and then etched in a solution of 10% oxalic acid and 90% distilled water, under a voltage of 12 V, for 40 s for microscopic characterization. During hydrogen charging, the tensile samples were sealed with epoxy resin except for an 18 mm gauge part (with an area of ~90 cm^2^) that was left exposed. Then, the tensile specimens were cathodically pre-charged in a 3.5% NaCl solution for 4 h at room temperature with a direct electron current density of 100 mA/cm^2^. The pH of the 3.5% NaCl solution was ~7 and the cathodic pre-charge potential was about −2.2 V. Figure 1b presents a diagram of the hydrogen charging experimental apparatus. After electrochemical hydrogen charging, tensile tests were undertaken using an Instron 5982 testing machine at room temperature with a low strain rate of 1 × 10^−5^/s. To ensure the accuracy and repeatability of the experiment, three tensile samples were tested for each state. The $I_{HE}$ is the HE-susceptibility index, which was evaluated according to the following equation [20]:(1)$$I_{HE}=\frac{\delta_{0}-\delta_{H}}{\delta_{0}}\times 100\%$$
where $\delta_{0}$ and $\delta_{H}$ are the elongations of samples with and without H, respectively.

### 2.3. Microstructure Characterizations

The overall fracture surfaces were observed using JSM-6510 scanning electron microscopy (SEM) equipment. An FEI Apreo field emission SEM, equipped with an electron backscattered diffraction (EBSD) detector was employed to characterize the microstructure before and after hydrogen charging. For SEM observations, the samples were prepared using electrochemical corrosion with a voltage of 12 V in an electrolytic solution (oxalic acid–distilled water = 1:9) at room temperature for 40 s. For EBSD observation, the samples were electropolishing in a solution (perchloric acid–ethanol = 1:9) with a voltage of 23 V at −30 °C. For TEM observation, the thin foils were prepared using twin-jet electropolishing with a voltage of 12 V in a solution (perchloric acid–ethanol = 1:9) at 0 °C.

### 2.4. Hydrogen Distribution Analysis by TOF-SIMS

The distribution of hydrogen was investigated using time-of-flight secondary ion mass spectrometry (TOF-SIMS) equipment. The samples were 6 × 6 × 3 mm^3^ in size, electrochemically polished and then subjected to hydrogen charging for 4 h for the TOF-SIMS analysis. The TOF-SIMS experiment used a Cs ion source with a voltage of 2 keV and a beam current of 47 nA to analyze an area of the sputtered crater of size 300 × 300 μm^2^. The depth profiling for the hydrogen distribution of H in the test steel was carried out by alternately using TOF-SIMS analysis and Cs sputtering at 2 keV.

## 3. Results

### 3.1. Tensile Properties of Uncharged and H-Charged Samples

To assess the effect of hydrogen charging on the mechanical behavior of SLM 316L SS, the tensile engineering stress–strain curves were measured during the SSRTs for both the uncharged and H-charging samples, as shown in Figure 2a. In the H-charged conditions, a yield strength (σ_0.2_) of ~487 MPa, and an ultimate tensile strength (UTS) of ~552 MPa, were slightly lower than the H-free conditions with an σ_0_._2_ of ~488 MPa and UTS of ~572 MPa. Nevertheless, the ε_f_ of the H-charged 4 h sample was only 36%, which is significantly lower than the H-free sample which had a ductility of 60% (see Figure 2c). Thus, the HE-susceptibility index $I_{HE}$ of SLM 316L SS was about 40%, indicating that SLM 316L SS has a relatively higher HE sensitivity, in a high current density of 100 mA/cm^2^. Figure 2b shows the strain hardening rate–true strain curves of uncharged and H-charging samples. As can be seen from Figure 2b, the strain hardening capacity decreased slightly after hydrogen charging, and the strain hardening capacity of the uncharged sample was higher and longer than the sample after hydrogen charging. Figure 2d and Table 3 display the other previously reported HE-susceptibility index of conventionally manufactured 316L austenitic stainless steels, but to compare the results in a more meaningful way, we leave our analysis to the end. As is shown, austenitic stainless steel generally has good resistance to hydrogen embrittlement; however, the current SLM 316L SS does not exhibit superior HE resistance, which may be related to the heterogeneous microstructure caused by SLM production.

### 3.2. Microstructures of Samples without and with H

The SEM results in Figure 3 show the microstructures of uncharged and H-charging samples. In the H-free sample, the melt pool (MP) appears with a typical ‘fish scale’ morphology structure, which is the result of the SLM process [30]. During the process, the laser beam is focused on the metal powder, and a melt pool is formed under the laser focusing spot. Various sizes and shapes of MPs were formed due to the different solidification conditions [30,31,32]. The mean depths and width of MPs were measured as approximately 30 μm and 100 μm, respectively. As shown in Figure 3a–c, the semicircular MPs are composed of columnar grains and cellular structures. Figure 3c is a high-magnification SEM image of Figure 3b, which shows the relatively finer cellular structures. Moreover, a finer nanometer structure exists in the cellular structure as shown in Figure 3c, which is based on a higher cooling rate and could produce a finer sub-structure [33].

Figure 3d–f shows the surface morphology of the H-charging sample. The largest portion of the hydrogen-charged specimen has not been influenced by hydrogen, but the edge of the sample has obvious cracks along the MP boundary (MPB) and the crack opening increases as the location become closer to the sample edge (as seen in Figure 3d). A more detailed SEM image in Figure 3e suggests that the H-induced cracks were initiated at the specimen edge and then propagated toward the MPBs (as indicated by the yellow arrows in Figure 3e). Compared with the H-charging sample, the edge of the sample without H has no cracks (as seen in Figure 3a,b). Figure 3f is a high-magnification SEM image of Figure 3e, which shows the cellular structure has not been changed by hydrogen.

**Figure 3 materials-16-01741-f003:**
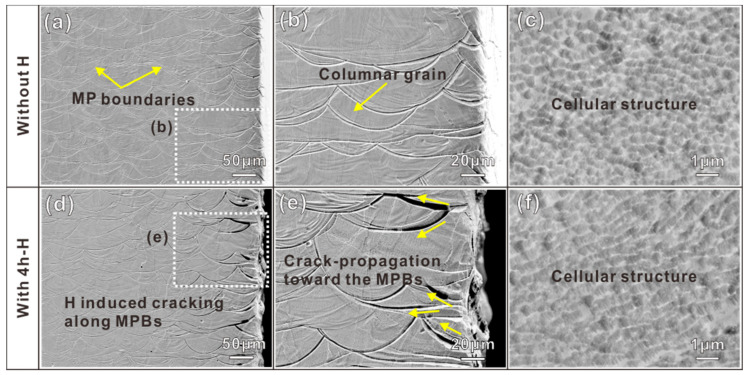
SEM maps of the surface of samples without (**a**–**c**) and with H-charged (**d**–**f**).

Figure 4 presents the quasi in situ EBSD observations [34] of the SLM 316L SS sample before and after hydrogen charging. Figure 4a,d presents the inverse pole figure (IPF) maps of the sample before and after H-charging, respectively. The typical heterogeneous MP structures and the columnar crystals inside the MPs showed little change after 4 h of hydrogen charging. The EBSD phase maps, with austenite in blue and martensite in red, are shown in Figure 4b,e, which indicates that both the uncharged and H-charging samples were mainly in the austenite phase (>99%), while the martensite phase (<1%) was sparsely distributed as red spots near the grain boundaries, which may represent the noise generated in the EBSD analysis. Thus, there was little phase transition after 4 h of hydrogen charging at 100 mA/cm^2^. However, both maps show that there is a large portion of low-angle grain boundaries (LAGBs, yellow line) with 5°–15° misorientation, but only a small portion of high-angle grain boundaries (HAGBs, black line) with >15° misorientation. Figure 4c,f shows the misorientation distribution plots of the uncharged and H-charging samples, respectively. Figure 4b,c,e,f, shows that the percentages of LAGBs increased after the 4 h hydrogen charging (76%), which indicates that hydrogen-induced lattice distortion contributed to the growth in LAGBs.

### 3.3. Fracture Analysis

Figure 5a,d displays the overall fracture morphologies of the uncharged and H-charging samples, respectively. The central region of the fracture surfaces (Figure 5b,e) is almost entirely composed of tiny dimples with a mean diameter of several hundred nanometers. In the peripheral fracture region (Figure 5c,f), we can observe many fine and shallow dimples in both the uncharged and H-charging samples. Interestingly, we found that the dimple sizes of the samples are in the submicron range. By comparison with the H-charging sample, there was a smaller and denser dimple morphology on the fracture surface of the uncharged samples which also suggested higher plasticity. Furthermore, there was no quasi-cleavage and intergranular feature in either sample, indicating that they were in ductile fractures mode with a significant number of dimples [15,35] and further indicating that the hydrogen charging in this study did not change the basic failure modes.

To study the effect of hydrogen charging on the loss of plasticity, the lateral surfaces of the tensile fractured samples under hydrogen-free and 4 h H charged conditions were observed, as shown in Figure 6a–d. As shown in Figure 6a,c, there was a large necking area in the hydrogen-free sample and the MPBs elongated along the tensile direction, which was vertical. The direction of extension in the MPBs, together with some fine cracks, is marked by the yellow and white arrows in Figure 6b,c, respectively. In the sample under the 4 h hydrogen-charging condition, the necking phenomenon is reduced, and some H-induced cracks (marked by the white arrows in Figure 6c) can be observed. In addition, the enlarged SEM micrograph showed that the tensile cracks were mainly propagated along the MPBs of the H-charging sample (Figure 6d). 

The fracture morphology confirmed ductility loss in SLM 316L SS after hydrogen charging. Figure 7 displays the EBSD maps of the microstructural changes on the lateral surfaces of the fractured tensile of the uncharged and H-charging samples. After tensile tests, deformation twins (as indicated by the red lines in Figure 7a,d) can be observed in both samples. As is evident from the post-deformation EBSD observations, the uncharged sample showed a higher density of twins when compared to the H-charging sample, which indicates a higher plasticity for the uncharged sample. Figure 7b,e shows the IPF map of both samples and, as a result of significant deformation, deformation, a certain number of deformation twins emerged in different orientations. The kernel average misorientation (KAM) images of the two kinds of samples are shown in Figure 7c,f, indicating that strain localization is generally concentrated at the MPBs of both samples. Additionally, there was a relatively low strain concentration at the twin boundaries and the deformation twinning could affect the work-hardening and ductility by blocking dislocation motion [36]. 

After the tensile tests, TEM characterizations of the uncharged and H-charging samples were carried out to further analyze the deformation mechanisms. The TEM samples were cut from the deformation area near the fracture surface of the tensile specimen. Figure 8a,b represents the TEM images that show many tangled dislocations and deformation twins with widths of about 20 nm in the uncharged sample. The white arrow in Figure 8b indicates a grain boundary (GB) interacting with different deformation twins on both sides, which implies that a significant number of deformation twins nucleated at the GBs. However, it is clear from the qualitative analysis that, in the 4 h H charging sample (Figure 8c,d), the dislocation density increased, and the density of the deformation twins decreased substantially when compared with the uncharged samples. The average twin spacing, and density of deformation twins, are much less than those of the as-received sample. The TEM results also indicate that dislocation sliding and deformation twinning are the dominant deformation mechanisms of the uncharged and H-charging samples.

### 3.4. TOF-SIMS Results Analysis

TOF-SIMS can directly detect the distribution of hydrogen atoms without chemical reactions. To further reveal the HE mechanism, TOF-SIMS was used to measure the local hydrogen concentration in SLM 316L SS with 4 h H charging. Figure 9a,b presents the EBSD and TOF-SIMS (area of the white box in Figure 9a) results, respectively. According to the TOF-SIMS results, the bright spots of H were mainly distributed along the MPBs (marked by white detached lines), which means that MPBs have a higher concentration of H. It also means that hydrogen atoms are more likely to be trapped at the MPBs after electrochemical hydrogen charging. The SIMS depth profiling results from the H^−^ signals are shown in Figure 9c. After electrochemical hydrogen charging at a constant cathodic current density value of 100 mA/cm^2^ for 4 h, the signal intensity of H^−^ is mainly on the surface of the sample. This result also means deeper penetration of the steel by hydrogen diffusion from the surface. Since the essence of electrochemical hydrogen charging is the diffusion of hydrogen from an aqueous solution to the hydrogen-filled surface, this is the surface of the H-charging sample is the main area affected by hydrogen. The hydrogen concentration decreases with increasing distance from the charged surface (see Figure 9c).

## 4. Discussion

### 4.1. Effect of Hydrogen on Mechanical Properties

In this study, the SLM 316L SS sample showed a significant decrease in elongation and a slight decrease in YS and UTS after 4 h of hydrogen charging, which is shown by the SSRT results in Figure 2. The mechanical properties are mainly decided by the microstructures. Previous studies have proved that hydrogen can facilitate the α′-martensite transformation and the presence of α′-martensite increases the hydrogen diffusion depth and HE sensitivity of the steels [37,38]. According to the EBSD results from the uncharged and H-charging samples (see Figure 4), martensite phase transformation was not observed after 4 h of hydrogen charging. This indicates the strong austenite phase of stability in 316L SS manufactured by SLM and a source of resistance to hydrogen embrittlement in a hydrogen environment. Hong et al. [39] reported that SLM 316L contains a unique cellular structure and high dislocation density, which increases the stability of austenite and reduces the production of α′ martensite. It also means that phase transition is not the reason for the loss of ductility in SLM 316L SS. The possible reason for the loss of ductility may be due to the reduced deformation twins formed in the H samples. During the plastic deformation, the samples with and without H both appear with high-density dislocations and some deformation twins. The twins can help to maintain strain hardening at higher stress levels, resulting in higher ductility [40]. However, the sample with H did not form high-density twins, which could be an important reason for the reduction in UTS and ductility of the samples with H. The surface damage induced by hydrogen will cause mechanical property degradation [14]. As shown in the SEM pictures of the hydrogen charging sample (Figure 3d,e), the hydrogen-induced cracking along the MPBs, which will be conducive to generating high-stress concentration, resulting in faster failure. 

### 4.2. H-Induced MPB Cracking

Compared with conventionally manufactured 316L austenitic stainless steels, the microstructure of SLM 316L SS has important features related to size. For example, semicircular MPs with a size of about 100 μm and cellular structures with a size of about 1 μm. These typical microstructures enable the SLM 316L SS to exhibit different HE behaviors. Figure 3d,e shows SEM pictures of the hydrogen charging sample and hydrogen-induced cracking can be seen along the MPBs. This could be an important reason for the reduction in ductility. Due to the rapid heating and cooling rate (10^3^–10^6^ °C/s) of the SLM process, it is easy to produce a large thermal gradient in the molten metal, which leads to a high concentration of elements along the MPBs [41,42]. Recently, some researchers reported that the hydrogen concentration regions in SLM specimens also have a higher concentration of chromium and manganese along the edges of the MPs [17,43]. In this study, it was observed that the hydrogen-induced cracks were propagated along the MPBs (Figure 3 and Figure 6), indicating that they act as hydrogen traps, resulting in higher hydrogen diffusion, higher concentration, and lower resistance to crack growth during hydrogen charging. The SIMS method effectively provides hydrogen distribution and depth profile information in the alloys [44,45]. As shown in Figure 9, the amount of hydrogen is enriched in MPBs after 4 h of hydrogen charging. To further illustrate the H-induced cracks in MPBs, Figure 10 provides the H-induced crack growth routes of the MP in the sample after H-charging and the application of strain. After 4 h of H-charging, some fine cracks gradually appear near the MPBs, and the hydrogen concentration crack tip (see Figure 10a,c). Under a slow strain rate of 10^−5^ s^−1^ tensile loading conditions, H-assisted cracks first appear in the edges of the melt pool boundaries, and then, under stress-driven H diffusion, hydrogen cracking moves along the MPBs and is then diffused to the next adjacent molten pool, finally leading to the fracturing of the sample (see Figure 10b,d).

## 5. Conclusions

This study concentrates on the hydrogen embrittlement (HE) behavior of an SLM additive manufactured 316L SS with specific melt pool microstructures. The main conclusions are summarized as follows:Hydrogen pre-charging caused cracking along the MPBs. Therefore, stress concentrations occurred around the cracks, which eventually led to the degradation of mechanical properties. In particular, tensile ductility decreased significantly from 60% to 36% in samples after 4 h of hydrogen charging.The hydrogen level rose at the MPBs of the SLM 316L SS sample after H-charging. This was mostly distributed on the surface of the sample, and the MPBs acted as hydrogen traps during hydrogen charging.The γ (austenite) to α’ (martensite) phase transition in the SLM 316L SS did not occur during hydrogen charging at room temperature because of the high austenite stability, which suggests that the martensite transition is not the reason for the reduction in the mechanical properties of the samples.Under a slow strain rate of 10^−5^ s^−1^ tensile loading conditions, H-assisted cracks first appeared on the sample MPBs’ surface, and then H diffused mainly along the MPBs, finally leading to the fracture of the sample.

## Figures and Tables

**Figure 1 materials-16-01741-f001:**
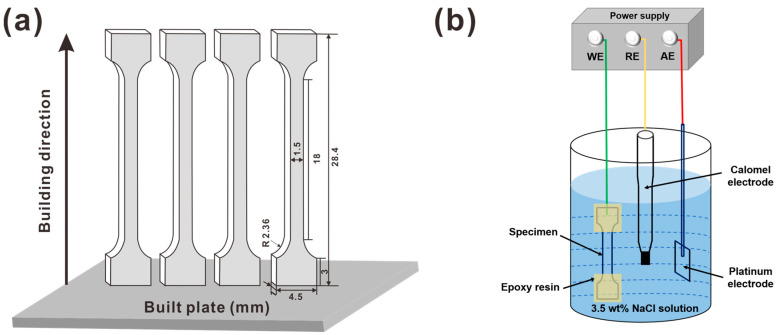
Diagram of (**a**) tensile sample, (**b**) the cathodic hydrogen charging process.

**Figure 2 materials-16-01741-f002:**
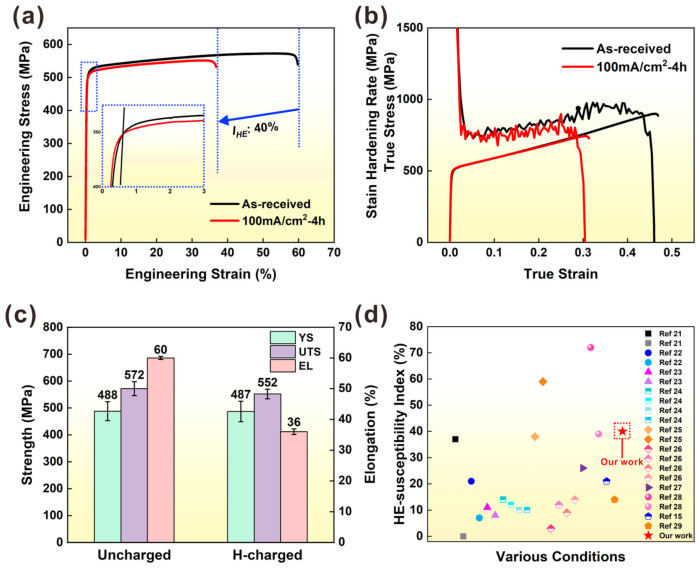
(**a**) Engineering tensile stress and strain curve of uncharged and H-charged samples; (**b**) the strain-hardening curves of uncharged and H-charged samples; (**c**) Tensile properties of uncharged and H-charged samples; (**d**) Comparison of HE susceptibility between this work and the reported studies for various 316L SS.

**Figure 4 materials-16-01741-f004:**
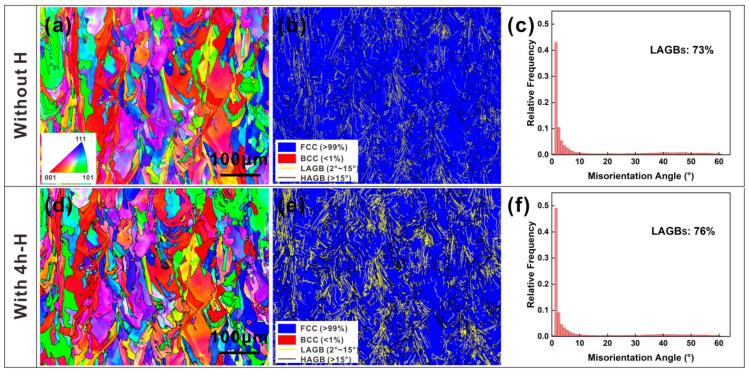
Quasi in situ EBSD maps of (**a**–**c**) uncharged sample and (**d**–**f**) H-charged sample. (**a**,**d**) inverse pole figure (IPF) grain orientation maps, (**b**,**e**) phase maps with the LAGBs and HAGBs, (**c**,**f**) misorientation angle distribution maps.

**Figure 5 materials-16-01741-f005:**
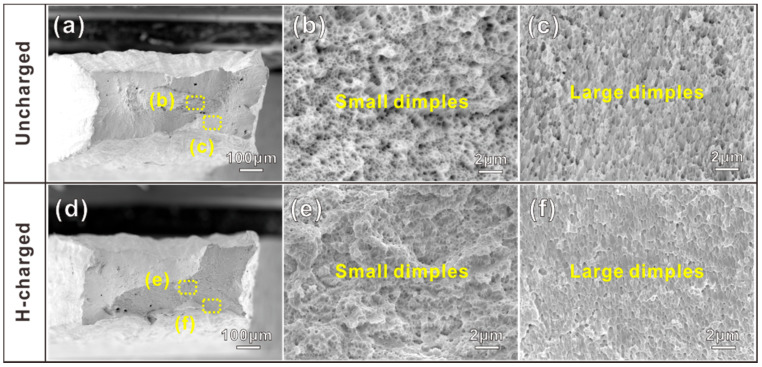
Tensile fracture surfaces of (**a**–**c**) uncharged sample; (**d**–**f**) H-charged sample. (**b**–**c**) and (**e**–**f**) are the detailed SEM images of the dotted squares in (**a**) and (**d**), respectively.

**Figure 6 materials-16-01741-f006:**
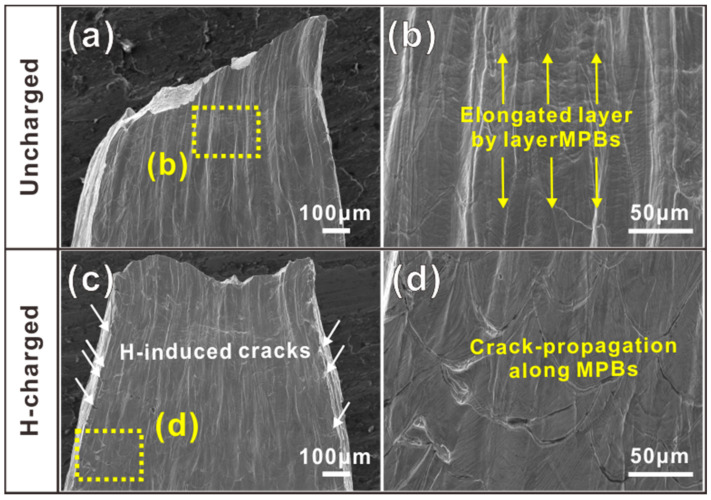
Tensile lateral surfaces of fractured samples of (**a**,**b**) uncharged sample and (**c**,**d**) H-charging sample.

**Figure 7 materials-16-01741-f007:**
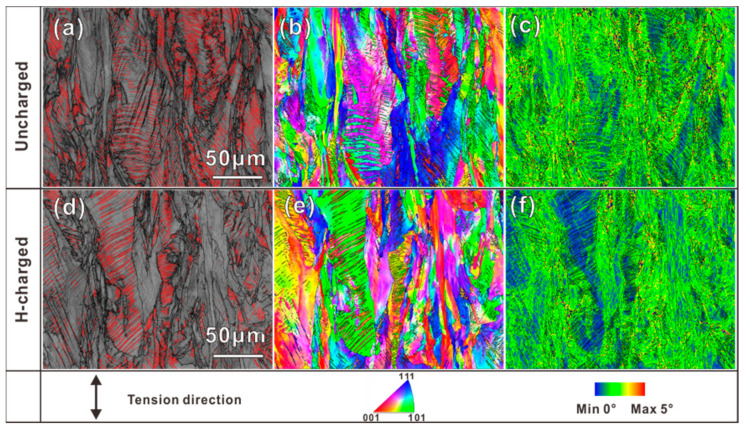
EBSD maps of the lateral fracture surfaces of (**a**–**c**) uncharged sample and (**d**–**f**) H-charging sample. (**a**,**d**) image quality (IQ) maps, (**b**,**e**) IPF orientation maps, (**c**,**f**) kernel average misorientation (KAM) maps.

**Figure 8 materials-16-01741-f008:**
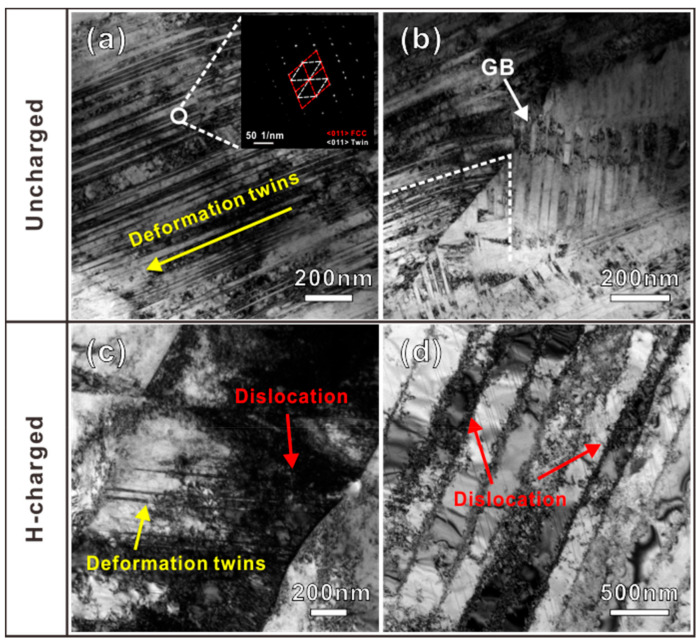
TEM images of the tensile deformation microstructures of (**a**,**b**) uncharged sample and (**c**,**d**) H-charging sample.

**Figure 9 materials-16-01741-f009:**
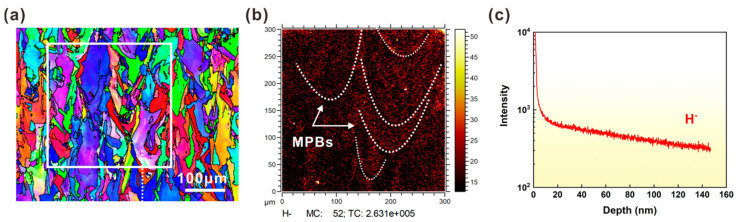
(**a**,**b**) EBSD and SIMS maps of the H-charging sample. The white square in (**a**) indicates the area of the SIMS map. (**c**) SIMS depth profiles of H^−^ in the H-charging sample show a rapid decrease in the tread with the distance from the sample surface.

**Figure 10 materials-16-01741-f010:**
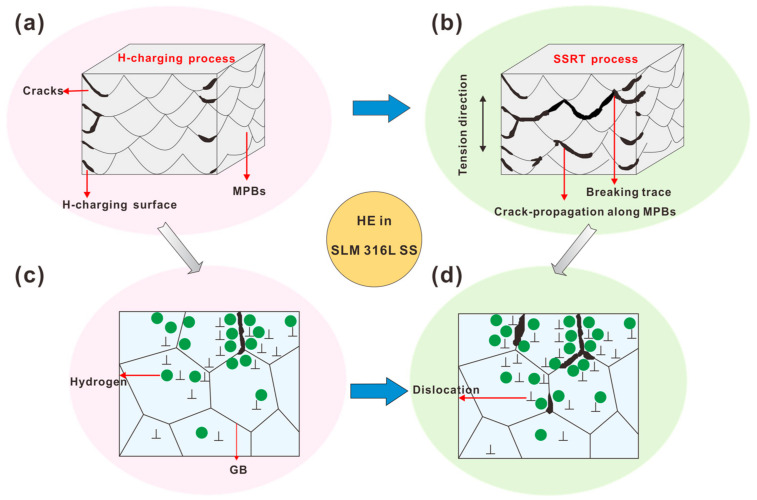
Schematic illustration of the crack growth routes along the MPBs with (**a**,**c**) H-charging process and (**b**,**d**) SSRT process.

**Table 1 materials-16-01741-t001:** Processing parameter of samples.

Laser Power/W	Scanning Speed/mm s^−1^	Hatch Spacing/μm	Layer Thickness/μm
**370.**	1300	80	30

**Table 2 materials-16-01741-t002:** Chemical composition of 316L SS powder (wt%).

Cr	Ni	Mo	Mn	Si	C	N	Fe
17.0	11.3	2.63	0.86	0.49	0.013	0.09	balance

**Table 3 materials-16-01741-t003:** Various conditions studies on HE resistance of 316L stainless steel.

Sample Details	Environment	H-Charging Condition	$I_{HE}\;(\%)$	Ref.
type-316L ODS-D5	gaseous hydrogen	10 MPa pressure	37	[21]
type-316L ODS-D150			0	
non-laser peened 316L	1 mol/L of H_2_SO_4_ + 0.125 g/L Na_4_P_2_O_7_ · 10H_2_O	50 mA/cm^2^ (24 h)	21	[22]
laser peened 316L			7	
Not homogenized 316L	0.5 M H_2_SO_4_ + 250 mg/L As_2_O_3_	200 A/m^2^ (50 min)	11	[23]
Homogenized 316L			8	
non-laser peened 316L	0.5 mol/L of H_2_SO_4_ + 1 g/L Na_4_P_2_O_7_ · 10H_2_O	20 mA/cm^2^ (96 h)	14	[24]
laser peened 316L (6 GW/cm^2^)			12	
laser peened 316L (8 GW/cm^2^)			10	
laser peened 316L (10 GW/cm^2^)			10	
hot-rolled 316L	electrochemical pre-charged	--	38	[25]
hot-rolled 316L	gaseous hydrogen	10 MPa pressure	59	
commercial type 316L	0.5 M H_2_SO_4_	4.5 mA/cm^2^ (12 h)	3	[26]
		4.5 mA/cm^2^ (24 h)	002012	
		4.5 mA/cm^2^ (36 h)	9	
		4.5 mA/cm^2^ (48 h)	14	
cold rolled 316L	0.5 mol/L H_2_SO_4_ + 1 g/L CH_4_N_2_S	20 mA/cm^2^ (18 h)	26	[27]
Types 316L (at −40 °C)	gaseous hydrogen	102 MPa pressure	72	[28]
Types 316L (RT)	gaseous hydrogen	83 MPa pressure	39	
DED 316L	gaseous hydrogen	120 MPa pressure	21	[15]
AISI 316L	0.5 M H_2_SO_4_ + 250 mg/L As_2_O_3_	400 A/m^2^ (50 min)	14	[29]

Abbreviations: ODS: oxide dispersed strengthened; D5: particle sizes 5 μm; D150: particle sizes 150 μm; DED: directed energy deposition; RT: room temperature; AISI: American Iron and Steel Institute.

## Data Availability

The data used in this study are available from the corresponding authors upon request.

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
