# Peer review of "Significance of Melt Pool Structure on the Hydrogen Embrittlement Behavior of a Selective Laser-Melted 316L Austenitic Stainless Steel"

_materials, 2023, doi:10.3390/ma16041741_

Round 1
Reviewer 1 Report
This is an interesting paper. However, the following comments and suggestions must be taken into account:
Question: On the SSRT tests, what was the criteria to use a strain rate of 1x10-5 s-1 instead a strain rate of 1x10-6 s-1 ?
Comment and question: For the electrochemical hydrogen charging tests, why NaCl electrolyte was chosen (see Table 3) ? What was the pH solution ? What was the cathodic pre-charging potential ?
Comment: Page 5, the text on lines 4-6 and Fig. 2(c) indicates yield strength values of 487 MPa and 492 MPa for uncharged and H-charged conditions, respectively. However, in both cases the curves in Fig 2 (a) seems to show values of about 500 MPa or a little higher. Could authors comment on this observation?....Perhaps a window from 450 MPa to 550 MPa may help to clarified this point.
Comment: The legend in Fig 2( c ) reads: …(b) Ture….Should red: ….(b) True
Comment: The first column in Table 3 reads: Sampleb Details….Should read: Sample Details
Page 5, Section 3.2 reads: Microstructural of samples….Should read: Microstructures of samples…..
Page 6, Fig. 3. Comment: Fig 3(a) is Ok. However, for the sake of comparison with Fig 3(c) , it would be possible to replace this image for other showing the edge of the sample without H ?
Page 6, line 18 reads: …. before and after hydrogen charging without any treatment. ….Comment: what kind of treatment authors are talking about ?
Page 6, lines 21-22 reads: The EBSD phase maps with austenite in blue and martensite in red are shown in Figs. 4 (b) and 4 (d),…..Comment: How can this statement be reconciled with the text in page 11, lines 8-12 and Conclusion 2 ?? (page 12).
Page 6, line 24 reads: Moreover, Figs. 4 (b) and 4 (c) shows the misorientation distribution maps of……Should read: Moreover, Figs. 4 (b) and 4 (d) shows the misorientation distribution maps of….
Page 8, line 7 reads: …..hydrogen charging condistion, Should read:..hydrogen charging condition,
Page 8, line 8 reads: …is obviously reducd, Should read:.. is obviously reduced,
Page 8, Fig. 6.- Fig. 6 is Ok. However, in order to improve the quality of Fig. 6(c), the color of text (legend) and arrows inside this figure should be yelow or white (red color gives a poor contrast).
Page 9, line 7 reads: … of twins which imples ….Should read:.. of twins which implies…..
Page 9.- The first paragraph on page 9 describes the various conditions of Fig. 7. However, nothing is mentioned on the meaning of Figs 7(b) and 7(e). Please check this and rewrite the sentence. Otherwise, Figs 7(b) and 7(e) needs to be removed.
Author Response
Please see the attachment, thanks a lot!

Reviewer 2 Report
Review
The submitted manuscript is interesting and contains new valuable results. However, the manuscript should be improved before publication in the journal according to the following comments.
1. The aim of the article should be clarified in Introduction.
2. In the text there are many typos (Pages 4 and 8; capture of Figures 2b and 9; subtitle 3.2). The manuscript should be checked and corrected by the authors.
3. Error bars should be added in Figure 2c.
4. All new abbreviations should be clarified below Table 3.
5. Was the marked cellular structure observed after H-charging (Figure 3)?
6. In Figure 4, the authors showed the misorientation distribution maps of the uncharged and H-charged samples. Why samples? It looks like the same sample before and after H-charging. Please, check and correct.
7. In the text was said that the percentages of LAGBs clearly increased in sample after 4 h hydrogen charging. Please, add quantitative estimation of the changes.
8. Tensile fracture surfaces were presented in Figure 5. It is not clear what is fracture at pre-cracked arias? Please, clarify it.
9. It is not clear how dislocation density, twin density and spacing of deformation twins were estimated after tension. Please, clarify it.
10. It would be better to show MPBs in SIMS map by detached lines (Figure 9b).
Author Response

(The authors gave the same response as above.)

Reviewer 3 Report
1) Authors must give a space before the reference number. For example "mechanical properties, corrosion resistance, biocompatibility and low cost[8, 9]", But it should be "mechanical properties, corrosion resistance, biocompatibility and low cost [8, 9]".
2) The introduction lacks the citations of similar research articles published recently. It is recommended to construct a table comparing the reported results with the present article's.
3) What is the reason for selecting specifically a 3.5% NaCl solution as an electrolyte? What happens when the electrolyte concentration is varied.
4) The line, To ensure the accuracy of the experiment, an extensometer was used and three parallel samples were tested for each state. What are the 3 parallel samples, brief them properly in the revised manuscript, Because this will create confusion to the readers.
5) Authors mentioned that after hydrogen embrittlement, the UTS is reduced. But not clearly discussed what is the mechanism and the reason.
6) Conclusion must contain the novelty of the work and the important findings. It is recommended to revise it accordingly.
Author Response

(The authors gave the same response as above.)
